# ReCell: replicating recurrent cell for auto-regressive pose generation

Vladislav Korzun
korzun@phystech.edu
Moscow Institute of Physics and
Technology
Moscow, Russia
Tinkoff
Moscow, Russia

Anna Beloborodova
beloborodova.as@phystech.edu
Moscow Institute of Physics and
Technology
Moscow, Russia
Tinkoff
Moscow, Russia

Arkady Ilin
arkady.ilin@skoltech.ru
Skolkovo Institute of Science and
Technology
Moscow, Russia
Tinkoff
Moscow, Russia

## ABSTRACT

This paper describes FineMotion's gesture generating system entry for the GENEA Challenge 2022. Our system is based on auto-regressive approach imitating recurrent cell. Combined with a special windowed auto-encoder and training approach this system generates plausible gestures appropriate to input speech.

## CCS CONCEPTS

• **Computer systems organization** → **Embedded systems**; *Redundancy*; Robotics; • **Networks** → Network reliability.

## KEYWORDS

embodied agents, neural networks, gesture generation, social robotics, deep learning

**ACM Reference Format:**
Vladislav Korzun, Anna Beloborodova, and Arkady Ilin. 2022. ReCell: replicating recurrent cell for auto-regressive pose generation . In *INTERNATIONAL CONFERENCE ON MULTIMODAL INTERACTION (ICMI '22 Companion), November 7–11, 2022, Bengaluru, India.* ACM, New York, NY, USA, 4 pages. https://doi.org/10.1145/3536220.3558801

## 1 INTRODUCTION

Automatic animation generation for humanoid characters is becoming increasingly popular. Most 3D characters are still animated using expensive motion capture or a small set of predefined animations.

Recent advances in facial animation generation [1, 4] provide a realistic facial expression from audio. Moreover, JALI Viseme Model [2] was already used in recent projects like Cyberpunk2077. Automatic gesture generation, on the other hand, has several existing approaches, including [3, 6, 7, 10], but no standard, ready-for-production solution. One possible explanation for that is the lack of a unified method for reliable comparison. Multiple standards for humanoid rigs and skeletons are also present making it even more challenging to use open-source solutions. Furthermore, some existing approaches generate motions in 3D, while others generate motions in 2D. As a result, comparing them is difficult.

The GENEA Challenge 2020 [8] was held to address this issue. The main goal of this challenge was to determine which models for automatic gesture generation based on speech recording performed better on the same data. The main result of this challenge was that some models outperformed the previous year's best methods results while remaining far from real motion. Even though the target domain was relatively simple: the dataset only contains recordings for one person, and the generated motion was examined for the upper body without fingers. Because there are numerous ways to improve the challenge target, the organizers decided to hold another challenge.

In comparison to the previous one, the task of GENEA Challenge 2022 is more difficult. The proposed models should generate motions for more than one person during the conversation. A person's behavior during a dialogue differs greatly from that of a single performance. The training and testing datasets included examples of the target person gesticulating while listening to the companion. However, these gestures are independent of speech. This behavior could have a significant impact on motion generation systems based on input speech.

We provide a solution that is based on both the speech input and the history of previously generated motions. This system generates plausible gestures from speech using a special windowed auto-encoder and training approach. We also tried to explain the difference between challenges and resulting models by testing one of the previous approaches that performed well on the previous challenge. Our code and some video examples are publicly available[1] to help other researchers reproduce our systems.

Our paper is organized as follows: Section 2 describes data processing, which is shared by all experiments; Section 3 describes our models; Section 4 discusses our results; and Section 5 is for the conclusion.

## 2 DATA PROCESSING

This time, the organizers of the challenge provided a large dataset based on the «Talking With Hands 16.2M» [9] dataset. It includes motion capture recordings as well as audio and textual transcripts of several people having a conversation. Mocap data includes motion data for the entire body, including fingers.

---

[1] https://github.com/FineMotion/GENEA_2022

Exploring the data we found several peculiarities which could affect the data-processing pipeline:

- Mocap skeleton differs from common industry formats (for example, Blender's Meta-Rig): the spine has a discontinuity, shoulders have extra bones and hands have a different wrist structure.
- Hand motion is inconsistent: some recordings lack data for finger motion, while others contain glitches and jerks.
- Audio recordings contain speech for both speakers. Some audio cuts are applied to the track in order to conceal confidential information, which greatly complicates speech processing.

To overcome the aforementioned difficulties, we attempted to construct the following data processing pipeline. The pose is represented by 3-dimensional axis-angle rotation vectors for all bone joints and also the position of the root bone. We also leave only non-constant values and normalize them over the training dataset's mean and maximum values. As the result, pose representations with 164 real value features are produced. We tried different representations for audio (MFCC, Mel-spectrogram, log Mel-spectrogram) with different framerate but found no significant difference during training. We chose MFCC extracted at the same framerate as motion (30fps).

As mentioned before, audio recordings contain speech for both speakers in dialogue. However, organizers provide transcripts only for the target person's speech. We assumed, that using audio while the target person is silent could help to predict visual response to the companion's speech. But to help model distinguishing the target person's speech from the other, we added the textual information as follows. First, we built a vocabulary of symbols on the training dataset. Second, we use one-hot vectors to represent symbols. Finally, we concatenate audio features with obtained vectors that are uniformly distributed within word spans. Figure 1 shows an example of such alignment.

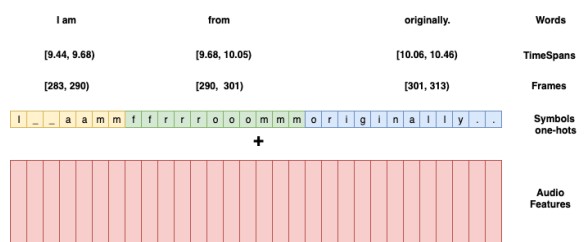

**Figure 1: Example of audio and text features alignment**

## 3 MODELS

### 3.1 Seq2seq

We started to design our architecture taking inspiration from a solution that performed well in the previous competition. The seq2seq model by [5] based on both audio and text features achieved the highest median score on human-likeness and the second highest on appropriateness. Unlike the previous work, we propose eliminating separate encoders for audio and text features. We also decided to use a simple seq2seq instead of a context encoder. To keep the

context encoder's main feature of using more context, we decided to use a wider input features window than the length of the output predicting sequence. Other components, such as the loss function and decoder initialization, remained unchanged. Figure 2 depicts the final seq2seq model scheme.

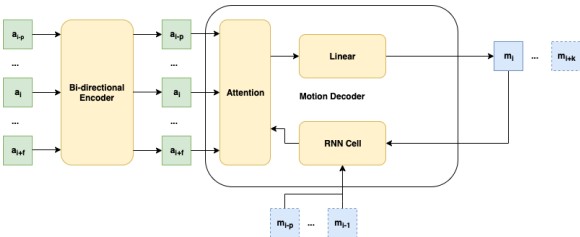

**Figure 2: Scheme of Seq2Seq model**

Here $a_{i-p}, \ldots, a_{i+f}$ is an input features window for output motion $m_i, \ldots, m_{i+k}$ , where $f >= k$. $m_{i-p}, \ldots, m_{i-1}$ are the output poses from the previous step.

This model produced good loss function values during our tests, but the resulting motions converged to minor movements around the rest pose. We also tried some training strategies, such as feeding zeros instead of previous poses and adding additional dropouts inside the motion decoder to force the model to focus on input features. However, this shows only minor improvements.

We believe this defect in the method manifested itself due to the significant differences in the datasets used in this and previous challenges. The last year's challenge dataset only included one actor's performance. There were no long pauses in monologue, in contrast to the dialogue speech, and no audio artifacts, allowing the model to create a consistent mapping between speech and gestures. As a result, we require a new model to address the following issues:

(1) Predict expressive motions from actual speech
(2) Realistically continue motion in the absence of speech

We tried to solve the aforementioned problems with the following architecture.

### 3.2 ReCell

We discovered that the model provided in [6] generates a variety of poses but they are unstable. This model creates a single animation frame from a sliding window of audio features. We used auto-regressive input to stabilize predicted motions. We also tweaked the feature window encoder. The original model for frame $i$ takes input $a_{i-k/2}, \ldots, a_{i+k/2}$, passes it through a single-direction GRU, and takes the last output. We decided to take the $k/2$-th output from bidirectional GRU, which corresponds to the current frame. We found that this modification produced slightly more expressive motion, which could be seen in our supplementary materials. We called this model "ReCell" because it replicates recurrent cell behavior. Figure 3 depicts the model's architecture.

To avoid the seq2seq model's convergence to a rest pose, we should strike a balance between auto-regressive input and speech features. The authors of [7] faced the same problem. To overcome this, they forced the model to extract useful features from the speech input by pretraining the model without auto-regression. Then they

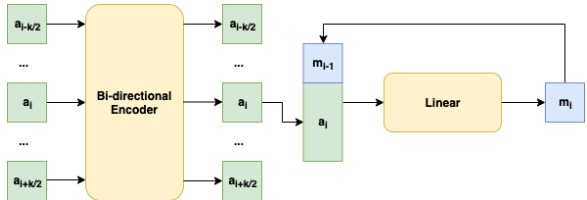

**Figure 3: ReCell model architecture**

also used the teacher forcing technique to let the model properly integrate auto-regressive input. As the result, we attempted to implement these ideas in the following manner.

In order to optimize the training procedure and allow teacher-forcing we first slice input data sequences into small sub-sequences of 10 successive poses and speech features each. On each training step, the model receives a batch of sub-sequences along with a pose corresponding to $(i-1)$-th element, where $i$ indicates the start of the sub-sequence. Then we iterate through this sub-sequence starting from $(i-1)$-th pose in an auto-regressive manner, passing the output of the previous iteration as an input in the current iteration.

We additionally apply small changes in training input depending on epoch number:

- During the first 10 epochs, we zero the starting pose as well as some poses between iterations. A number of epochs were chosen empirically, as the loss stopped decreasing after 10 epochs without zeroing.
- After the first 10 epochs, we use starting pose from data, but instead, replace some intermediate predicted poses with real ones imitating the teacher-forcing technique.

The precise strategy for replacing frames along with a description of model hyperparameters will be provided in 3.4. On prediction, we go through the entire recording frame by frame, beginning with zero pose as auto-regressive input.

### 3.3 Windowed auto-encoder

To overcome motion instability we trained a simple linear auto-encoder to reduce the dimension of the output vectors. It has two major components, as usual: encoder and decoder. Input poses are projected into lower-dimensional space by the encoder. In turn, the decoder must reconstruct the input pose from its lower-dimensional representation.

The encoder has one hidden layer, that converts the input vector to a size of 512. After that, we apply ReLU as an activation function. Then we apply another linear layer and Tanh function to normalize. We set the final size of the output vector equal to 60 as it is the smallest possible size that allows retaining enough information while also providing compact representation. When we decreased the size of the output vector we started to notice visible artifacts on the reconstructed motion.

We use one additional hidden linear layer to restore the vector from the encoded state, which translates the encoder output vector of dimension 60 into 512. Next, a ReLU function and an output linear layer were used, which translates the vector of 512 into dimension 164. Since normalized features with values that range between [0, 1] were fed to the auto-encoder input, the decoder output is then

normalized by the Tanh function and shifted to [0, 1], which showed better results than Sigmoid during our experiments.

We discovered that when such an auto-encoder is trained, the reconstructed pose is mostly similar to the input pose, but it occasionally provides unrealistic poses for the left shoulder. We believe this behavior is related to rare jerks in training data. To address this issue, we decided to improve our auto-encoder by reconstructing a short sliding window of poses rather than the single pose. As a result, we stack three sequential frames into a combined vector and fed it to the auto-encoder, implying that the initial and output vectors were of dimension $164 \times 3$. In everything else, the auto-encoder architecture is similar to that described above. In order to get a pose for the current frame we average its corresponding poses from all sequential windows it appears in (e.g for 5th pose we average pose vectors $p_5$ from sequences $[p_3, p_4, p_5], [p_4, p_5, p_6], [p_5, p_6, p_7]$).

### 3.4 Final pipeline

Our final approach is as follows. As the main model, we used the ReCell model with the following parameters.

First of all, our Bi-directional encoder consists of a linear highway and GRU. Linear highway, in turn, consists of three sequential blocks with linear layer, batch normalization, and dropout with ReLU as the activation function to encode audio features. As mentioned before, we use GRU to extract contextual features from the window of audio features. The size of this window is equal to 61. All hidden sizes are equal to 150.

At the same time, we put the pose from the previous step through another highway with a bottleneck of size 40. It contains only two linear layers with ReLU between them and dropout to prepare model output for auto-regression. It enables us to obtain more complex and informative features, as well as translate vectors to the required size. This operation we call Hidden Highway.

Finally, outputs of the Bi-directional encoder and Hidden Highway are concatenated together to create the final feature vector. Then, we apply another batch normalization, ReLU, and Dropout to the resulting feature vector and push it through the last linear layer to produce the next pose.

During the teacher-forcing, we zeroed frames between iterations with a probability of 0.5 in the first ten epochs, and we also replaced some frames with real ones with a probability of 0.5 in the following epochs.

To train the network, we use MSELoss and the Adam optimizer with default parameters from PyTorch.

We also train the main model to predict audio-encoder representations rather than poses. Then, using the Savitsky-Golay filter, we reconstruct the poses and smooth the predictions. We use 26 MFCC coefficients concatenated with one-hot encodings of text transcripts as input speech representation.

It's also worth noting that our auto-encoder was trained using the entire train dataset. However, in order to train the ReCell model we selected samples from the most frequently encountered speaker. We also removed audio cuts-related fragments.

## 4 RESULTS AND DISCUSSION

As in the previous challenge, the organizers provide [11] human-evaluation results for comparing the systems. As before there are

**Table 1: Summary statistics of user-study ratings**

| | Human-likeness | | Appropriateness |
| | | | Percent matched |
| ID | Median | Mean | (splitting ties) |
|---|---|---|---|
| FNA | 70 ∈ [69, 71] | 66.7 ± 1.2 | 74.0 ∈ [70.9, 76.9] |
| FBT | 27.5 ∈ [25, 30] | 30.5 ± 1.4 | 51.6 ∈ [48.2, 55.0] |
| FSA | 71 ∈ [70, 73] | 68.1 ± 1.4 | 57.1 ∈ [53.7, 60.4] |
| FSB | 30 ∈ [28, 31] | 32.5 ± 1.5 | 53.8 ∈ [50.4, 57.1] |
| FSC | 53 ∈ [51, 55] | 52.3 ± 1.4 | 53.0 ∈ [49.5, 56.3] |
| FSD | 34 ∈ [32, 36] | 35.1 ± 1.4 | 51.5 ∈ [48.1, 54.9] |
| FSF | 38 ∈ [35, 40] | 38.3 ± 1.6 | 51.7 ∈ [48.2, 55.1] |
| FSG | 38 ∈ [35, 40] | 38.6 ± 1.6 | 54.8 ∈ [51.4, 58.1] |
| FSH | 36 ∈ [33, 38] | 36.6 ± 1.4 | 60.5 ∈ [57.1, 63.8] |
| FSI | 46 ∈ [45, 48] | 46.2 ± 1.3 | 55.1 ∈ [51.7, 58.4] |

two main metrics: human likeness and appropriateness. The first evaluates how predicted motions are realistic, and the second estimates the correspondence between speech and predicted motions from it. While human-likeness evaluation follows the previous challenge, the approach to estimating appropriateness is different.

Investigating the results of the previous challenge organizers found that calculated appropriateness was highly correlated with the quality of generation motion. Therefore, this time they decided to calculate appropriateness in a different way. They ask evaluation study participants not to estimate the result videos alone but compare the corresponding audio-video pair with the mismatched one. It could help to find out if generated motion really appropriate to speech or just random. Table 1 presented by organizers shows the results of human-evaluation.

Our system (FSD) achieved slightly poor results on both metrics for the full-body motion: 34 on the human-likeness median score and 51.5% on Appropriateness percent matched.

Our model outperforms the provided baseline by human-likeness, although, there are models that showed significantly better results. Moreover, there is a model that outperforms even real motion by human-likeness. This leads us to assume that there is huge room for improvement in our approach.

The low score in human-likeness could be connected with insufficient fluidity of movement and periodic jerks. It could be smoothed with an even stronger filter but that could result in inexpressive, overly smoothed «jelly-like» movements. Also the lowest score of appropriateness means that generated motions do not have any semantics. It is only hand movements that follow the speech rhythm. Additional features, like Glove or BERT embeddings for a textual transcript, could be used to add semantics. We suppose that results can further be improved by extensive data cleanup. For example, only motions corresponding to actual speech can be used for training. Thus, even the seq2seq approach has room for improvement and presumably is able to show better results.

During the previous 2020 GENEA Challenge organizers also found that objective metrics such as average jerk and Hellinger distance may not correlate with visual quality. Therefore, we decided not to rely on objective metrics during the development of our system and to use only visual assessment. However, organizers

also provided [11] some objective metrics for the final evaluation of the submitted approaches.

## 5 CONCLUSION

The proposed challenge raised issues that are difficult to solve directly with a tried-and-true method. However, with additional data processing and training techniques, it may be possible to solve the problem relatively successfully. Although the proposed approach is far from ideal it has the potential to be improved. We believe that, with some modifications, the alternative sequence-to-sequence approach could also produce better results.

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
