# OpenReview forum: "ReCell: replicating recurrent cell for auto-regressive pose generation"
_ACM.org/ICMI/2022/Workshop/GENEA — GENEA Challenge & Workshop 2022 Workshopproceeding_

### Official Review · Reviewer_3Ybz · 2022-08-08
**I like the idea of using auto-regressive input, but how much did it improve the stability of prediction?**

**Rating:** 6
**Confidence:** 4

**Review:**

The paper explores two models to generate co-speech gesture. The one is seq2seq model proposed in previous research[5] and the other is new "ReCell" model that is based on a model proposed in [6].

Both models predict human pose or a sequence of poses as 3-dimensional axis-angle rotation vector from audio and text input.
The paper uses the MFCC feature as audio representation. The paper also uses text information where each character is represented as a one-hot vector.

The paper might need some work in organization. For example, In the section "3.2 ReCell" include some description about "seq2seq" model that should be written in the section "3.1 seq2seq".

The main contribution of the paper is proposed "ReCell" model that is modification of the previously proposed model with adding auto-regressive input. The change in proposed method is incremental.

I like the idea of adding auto-regressive input. I believe it could produce more time consistent predictions. The paper also stated it improved stability of the prediction. However, the paper did not provide any evidences such as quantitative analysis or sub-materials.

---

### Official Review · Reviewer_VXha · 2022-08-08
**Practical work but the novelty is unclear.**

**Rating:** 5
**Confidence:** 4

**Review:**

The authors propose an auto-regressive gesture generation model by inputting mel-frequency cepstral coefficients and character level word embeddings.

Strengths:
1. The authors start from a fundamental sequence-to-sequence model and provide their solutions to each problem they have encountered during the implementation: 1) apply auto-regressive generation to stabilize the output motions; 2) propose a training algorithm to make the model not converge to rest pose; 3) propose a windowed auto-encoder for the unrealistic poses for left shoulder.
2. The paper used character level word embeddings, unlike most works for gesture generation using text as input which use word level embeddings. Although the authors did not discuss much about this point, this could potentially be valuable for analyzing the difference between using character level embedding and word level embedding.

Weaknesses:
1. The novelty is unclear. This is due to mainly two reasons: 1) although the authors used techniques as mentioned in Strengths-1, they are not original (except Strengths-1-3). Strengths-1-(1,2) has been used in [7]. Although Strengths-1-3 is different from auto-encoder used in [6], the authors did not provide the comparison between these methods. 2) the author’s discussion for the proposed systems is not comprehensive. For instance, at line-163-164, while the authors wrote “During our tests, this modification produced better results”, it is unclear what is “better results”. Additionally, for the proposed windowed auto-encoder, no ablation was provided; thus, it is difficult to determine if or how much this component is useful.
2. Some details are missing. 1) At line-194, the authors used 10 epochs for zero-out. It is unknown that why it was 10 epochs. 2) For the auto-encoder, why the authors chose 60 as the final dimensionality.
3. The English writing requires efforts to understand. For instance, at line-194, the authors wrote “… as well as some frames between iterations.” The authors need to clarify what they are referring to with “between iterations”.
4. Figures that compare the proposed model and other systems has not been presented to better illustrate the results. Also, the results for objective evaluation were not presented.
5. No video was provided for examining the generated results.

---

### Official Review · Reviewer_ozsj · 2022-08-08
**ReCell: replicating recurrent cell for auto-regressive pose generation**

**Rating:** 5
**Confidence:** 5

**Review:**

# ReCell: replicating recurrent cell for auto-regressive pose generation

## Description

The paper describes an auto-regressive gesture generator, ReCell, for the GENEA Challenge 2022. The approach is trained and tested only using the provided data, which is modified from Lee G. et al. (2019) Talking with Hands. The work uses MFCC audio features, and a one-hot encoding of text symbols.

The authors develop their solution by first developing a seq2seq model, then resolving some short comings of that model, to arrive at the ReCell model.

The seq2seq model generated a single frame of motion for a sequence of audio features. The authors note the pose as unstable, so return the previous motion frame to concatenate with the current audio frame. Thus forming the auto-regressive model.

The motion data is compressed with an auto-encoder to provide an embedding of the rotation values.

## References

I suggest including the following work, which uses an auto-regressive approach:

Alexanderson S, et al. (2020). Style-Controllable Speech-Driven Gesture Synthesis Using Normalising Flows.

## Clarity of Exposition

I was not clear on how the authors incorporate the speaker identity on the model.
Line 83 mentions "distinguishing target speakers..." but only describes an encoding of text.

The authors do not describe their own evaluations of the model.
On line 141, they discuss some problems with convergence, but don't show this in any understandable way.

The lack of comparative evaluation is the main weakness of the paper, please include the evaluations from the GENEA challenge and use as part of the discussion.

## Reproducibility

The description of the method is quite high level. To enable reproduction please show a detailed description of the final model. I suggest replacing the figure of the seq2seq model that was not used.

## Conclusion

The paper describes an approach for gesture generation from speech features. The authors develop an earlier model, but do not show clearly what motivated the development. Too much of this paper describes a method that was not used. The paper also needs to include more comparative evaluation.

---

### Decision · Program_Chairs · 2022-08-11

**Decision:**

Accept (Workshop proceeding)

**Comment:**

This paper elicited a mixed response among reviewers, with two out of three reviewers favouring rejection. A strength of the work is that it clearly sets out to extend a prior approach in order to address its shortcomings, although this was also perceived as limited novelty by some. All reviewers remarked that the exposition was unclear, which also affects reproducibility. Reviewers also wanted to see more evidence of the positive effects of the proposed changes.

On balance, the chairs have decided to accept this paper to the GENEA Workshop, which means it will appear in the ACM ICMI adjunct proceedings, rather than the main proceedings.

We suggest that the authors carefully consider the feedback received from the reviewers and use it to improve their manuscript for the workshop camera-ready submission. Below follows some input from the chairs, based on the paper and the reviews:

1) The most important point when revising the paper for publication in the workshop is to perform modifications to address the reviewer’s comments on clarity and presentation.

2) It is also required to mention your team name in the paper, but we were unable to find it in the text. Please add it if it is not there. Alternatively, the organisers can change your team name to ReCell, but then you must let them know about that immediately.

3) Related to the above, it is permitted to not mention what condition ID one’s system had in the evaluation. However, the authors should still consider adding that information.

4) Perhaps the conclusion in the paper can be more specific regarding what the authors believe are the most important modifications to incorporate in order to further improve human-likeness in the future, based on what they have learnt from the challenge?

5) Another suggestion based on reviewer input would be to use the visualisation pipeline and stimuli shared with participating teams to conduct a new ablation study, in order to more directly study the results with and without the enhancements proposed in the paper, whilst keeping all other aspects fixed. This may or may not be feasible in the time that remains before the deadline. In the absence of that, perhaps video examples of models with and without these proposed enhancements can be uploaded, with a link to these videos added to the paper.